# Parallel evolution of reduced cancer risk and tumor suppressor duplications in *Xenarthra*

**Juan Manuel Vazquez[1], Maria T Pena[2], Baaqeyah Muhammad[3], Morgan Kraft[3], Linda B Adams[2], Vincent J Lynch[3]***

[1]Department of Integrative Biology, Valley Life Sciences, University of California, Berkeley, Berkeley, United States; [2]United States Department of Health and Human Services, Health Resources and Services Administration, Health Systems Bureau, National Hansen's Disease Program, Baton Rouge, United States; [3]Department of Biological Sciences, University at Buffalo, SUNY, Buffalo, United States

**Abstract** The risk of developing cancer is correlated with body size and lifespan within species, but there is no correlation between cancer and either body size or lifespan between species indicating that large, long-lived species have evolved enhanced cancer protection mechanisms. Previously we showed that several large bodied *Afrotherian* lineages evolved reduced intrinsic cancer risk, particularly elephants and their extinct relatives (*Proboscideans*), coincident with pervasive duplication of tumor suppressor genes (Vazquez and Lynch, 2021). Unexpectedly, we also found that *Xenarthrans* (sloths, armadillos, and anteaters) evolved very low intrinsic cancer risk. Here, we show that: (1) several *Xenarthran* lineages independently evolved large bodies, long lifespans, and reduced intrinsic cancer risk; (2) the reduced cancer risk in the stem lineages of *Xenarthra* and *Pilosa* coincided with bursts of tumor suppressor gene duplications; (3) cells from sloths proliferate extremely slowly while *Xenarthran* cells induce apoptosis at very low doses of DNA damaging agents; and (4) the prevalence of cancer is extremely low *Xenarthrans*, and cancer is nearly absent from armadillos. These data implicate the duplication of tumor suppressor genes in the evolution of remarkably large body sizes and decreased cancer risk in *Xenarthrans* and suggest they are a remarkably cancer-resistant group of mammals.

*For correspondence:
vjlynch@buffalo.edu

Competing interest: The authors declare that no competing interests exist.

## Editor's evaluation

This study is a useful extension of previous work on the relationship between body size and cancer risk and the mechanisms by which large-bodied mammals reduce their cancer risk. Through solid analyses of the genomes and several aspects of the cell biology of sloths, armadillos and their relatives, the study explores whether the evolution of large body size in their relatives (some extinct) was correlated with genomic changes such as the duplication of tumor suppressor genes, experimentally demonstrating that cells of Xenarthrans (sloths, armadillos, anteaters) are exceptionally sensitive to DNA damage. The study concerns a topic of great interest and contributes to our understanding of how cancer risk has evolved in mammals.

## Introduction

The evolution of large bodies and long lifespans in animals is constrained by an increased risk of developing cancer (*Caulin et al., 2015*; *Caulin and Maley, 2011*; *Nagy et al., 2007*; *Peto, 2015*). All cell types have an intrinsic risk of malignant transformation, and encode the same cancer suppression

mechanisms; thus, large organisms with many cells should have a proportionally higher risk of developing cancer than smaller organisms with fewer cells. In addition, the cells of organisms with long lifespans have more time to accumulate cancer-causing mutations and other types of damage than organisms with shorter lifespans and therefore should be at an increased risk of developing cancer. This risk is compounded by a general positive correlation between body size and lifespan across vertebrates (*Caulin et al., 2015*; *Caulin and Maley, 2011*; *Nagy et al., 2007*; *Peto, 2015*). While this strong positive correlation between body size, age, and cancer prevalence holds within species (*Dobson, 2013*; *Green et al., 2011*; *Nunney, 2018*), between mammalian species there is no correlation between either body size or lifespan and cancer risk (*Abegglen et al., 2015*; *Boddy et al., 2020*; *Bulls et al., 2022*; *Vincze et al., 2022*). This lack of correlation is often referred to as 'Peto's Paradox' (*Peto, 2015*).

Although many mechanisms can potentially resolve Peto's paradox, few have been studied experimentally, such as in *Drosophila* (*Shepherd et al., 1989*; *Parkes et al., 1998*; *Peleg et al., 2016*; *Garschall et al., 2017*), rodents (*Salmon et al., 2008*; *Seluanov et al., 2009*; *Liang et al., 2010*; *Gorbunova et al., 2012*; *Azpurua and Seluanov, 2013*; *Tian et al., 2019*; *Zhang et al., 2021*), bats (*Foley et al., 2018*; *Koh et al., 2019*; *Kacprzyk et al., 2021*), tortoises (*Glaberman et al., 2021*), and elephants (*Abegglen et al., 2015*; *Sulak et al., 2016*; *Vazquez et al., 2018*). We and others have shown that elephants, for example, evolved cells that are extremely sensitive to DNA damage (*Abegglen et al., 2015*; *Sulak et al., 2016*) at least in part through duplication of tumor suppressor genes (*Caulin et al., 2015*; *Sulak et al., 2016*; *Vazquez et al., 2018*; *Tollis et al., 2020*; *Vazquez and Lynch, 2021*). While this burst of tumor suppressor duplication occurred coincident with the evolution of reduced intrinsic cancer risk in *Proboscidea* (*Vazquez and Lynch, 2021*), we found that some other mammalian lineages such as in *Xenarthra* (armadillos, sloths, and anteaters) (*Figure 1A*) may also have evolved reduced intrinsic cancer risk and increased tumor suppressor dosage (*Vazquez and Lynch, 2021*). Interestingly, while living *Xenarthrans* are relatively small bodied, several lineages of recently extinct sloths (*Megatherium* and *Mylodon*) and armadillos (*Glyptodon*) independently evolved very large body sizes (*Figure 1B*; *Delsuc et al., 2016*). This suggests that at least some *Xenarthrans* have the developmental potential to be much larger bodied than extant species and thus must have evolved ways to reduce their intrinsic cancer risk.

Here, we reconstruct the evolution of body mass, lifespan, and intrinsic cancer risk across living and extinct *Xenarthrans* and show that intrinsic cancer risk decreased dramatically in the *Xenarthran* stem-lineage as well as several extinct lineages of giant armadillos and sloths. Using comparative genomics and phylogenetic methods, we found that these episodes of decreased cancer risk occurred coincident with the bursts of tumor suppressor gene duplication in the *Xenarthra* and *Pilosa* stem-lineages. Furthermore, we show that cells of these species are particularly sensitive to DNA damaging agents. Finally, we compare cancer prevalence across species using a new data set of pathology reports from a large colony nine-banded armadillos (*Dasypus novemcinctus*), finding that not only is the prevalence of cancer in *Xenarthrans* lower than most other mammalian lineages, but that armadillos have among the lowest cancer prevalence reported for any mammal. Similar to our previous study of body size evolution in *Proboscidea*, these data suggest that duplication of tumor suppressor genes occurred coincident with reductions in intrinsic cancer risk and likely facilitated the evolution of several extinct species with exceptionally large body sizes and low cancer risk.

## Results and discussion
### Evolution reduced cancer risk in *Xenarthra*

We previously used a data set of lifespan and body size data from over 1600 *Eutherian* mammals, focusing on *Afrotheria*, and used ancestral reconstruction to identify lineages to show that substantial accelerations in the rate of body mass and lifespan evolution occurred in multiple lineages, most notably in large bodied *Afrotherians* (*Figure 2*; *Vazquez and Lynch, 2021*). Unexpectedly, we also observed that several *Xenarthran* lineages also showed episodes of rapid body size and lifespan evolution suggesting this clade may also be a good representative in which to study the relationships between life history evolution and cancer risk (*Vazquez and Lynch, 2021*). To explore in greater detail the relationships between body size, lifespan, and intrinsic cancer risk in *Xenarthrans*, we assembled a time-calibrated supertree of *Eutherian* mammals by combining the time-calibrated molecular

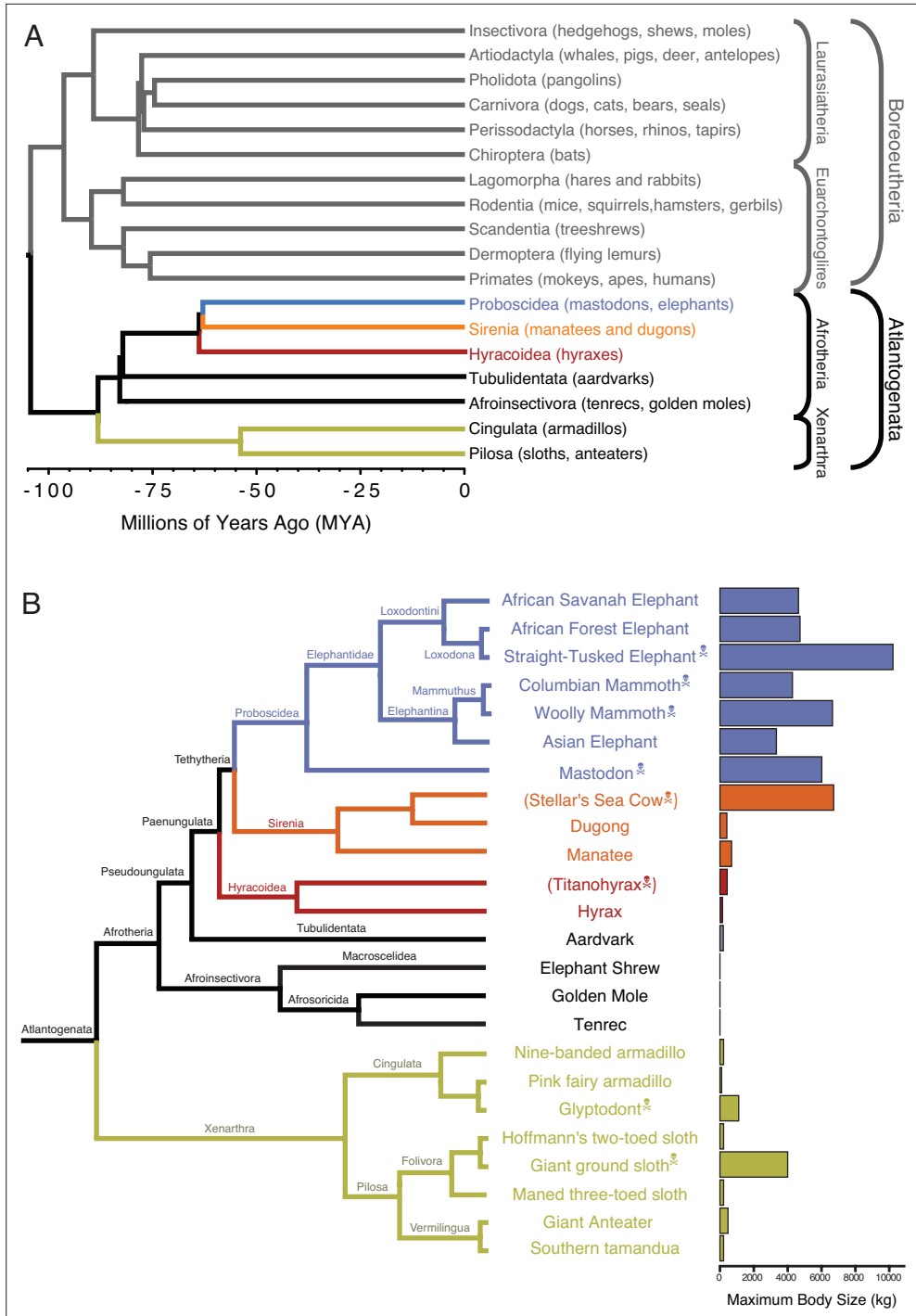

**Figure 1.** *Eutherian* phylogenetic relationships. (**A**) Phylogenetic relationships between *Eutherian* orders, examples of each order are given in parenthesis. Horizontal branch lengths are proportional to time since divergence between lineages (see scale, Millions of Ago [MYA]). The clades Atlantogenata and *Boreoeutheria* are indicated, the order *Proboscidea* is colored blue, Sirenia is colored orange, and *Hyracoidea* is colored red. (**B**) Phylogenetic relationships of extant and recently extinct *Atlantogenatans* with available genomes are shown along with clade names and maximum body sizes. Note that horizontal branch lengths are arbitrary, species indicated with skull, and crossbones are extinct and do not have available nuclear genomes.

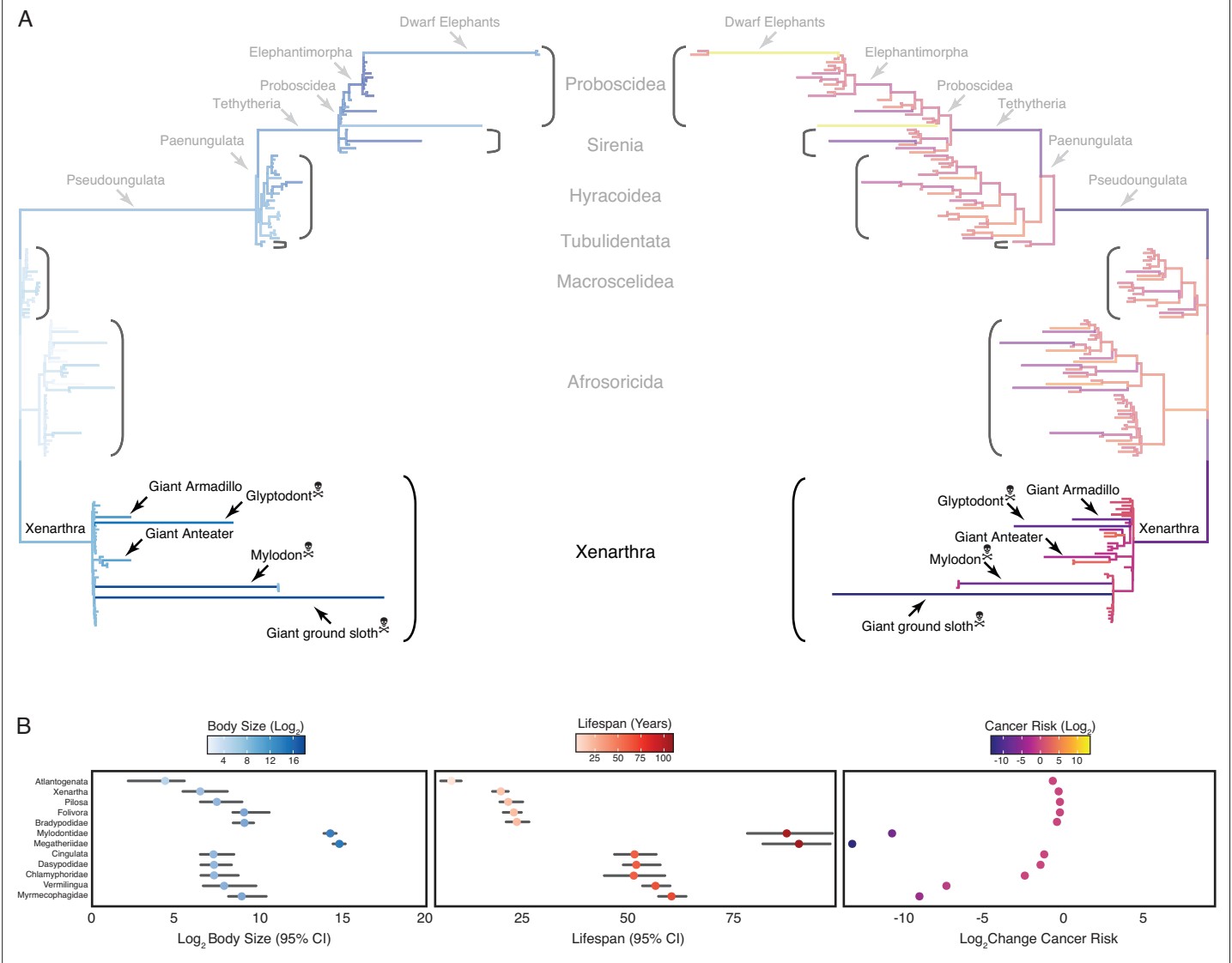

**Figure 2.** Convergent evolution of large-bodied, cancer-resistant *Xenarthrans*. (**A**) *Atlantogenata* phylogeny, with branch lengths scaled by $\log_2$ change in body size (left) or $\log_2$ change in intrinsic cancer risk (right). Branches are colored according to ancestral state reconstruction of body mass or estimated intrinsic cancer risk (scale shown in panel **B**). The *Afrotherian* part of the tree is shown opaque because it was analyzed in *Vazquez and Lynch, 2021*. (**B**) Ancestral reconstructions of body size (left), lifespan (middle), and change in intrinsic cancer risk (right). Data are shown as mean (dot) along with 95% confidence interval (CI, whiskers) for body size and lifespan (which is estimated from body size data).

phylogeny of *Bininda-Emonds et al., 2007* with the time-calibrated total evidence *Afrotherian* phylogeny from *Puttick and Thomas, 2015* and the time-calibrated *Xenarthran* phylogeny of *Delsuc et al., 2016*. While the *Bininda-Emonds et al., 2007* phylogeny includes 1679 species, no fossil data are included. The inclusion of fossil data from extinct species is essential to ensure that ancestral state reconstructions (ASRs) of body mass are not biased by only including extant species. This could lead to inaccurate reconstructions, for example, if lineages convergently evolved larger body masses from a small-bodied ancestor. In contrast, the total evidence *Afrotherian* phylogeny of *Puttick and Thomas, 2015* includes 77 extant species and fossil data from 39 extinct species, while the *Delsuc et al., 2016 Xenarthran* phylogeny is based on mitogenomes for 25 living and 12 extinct *Xenarthran* species that are broadly representative of the *Xenarthran* diversity. The combined supertree includes body size data for 1775 *Eutherian* mammals.

Next, we jointly estimated rates of body mass evolution and reconstructed ancestral states using the 'Stable Model' implemented in StableTraits (*Elliot and Mooers, 2014*), which allows for large

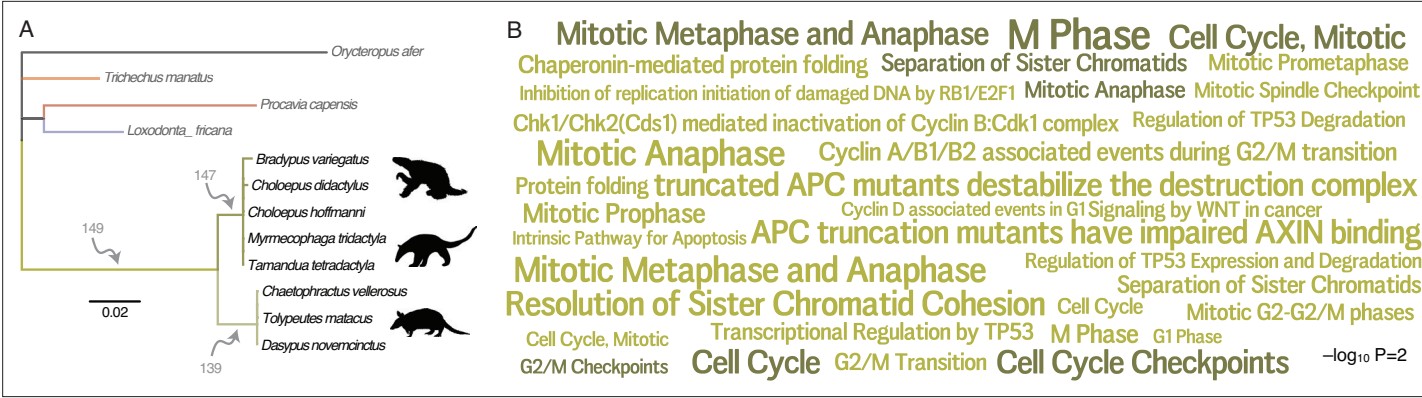

**Figure 3.** Pervasive duplication of tumor suppressors in *Xenarthra*. (**A**) Atlantogenata phylogeny indicating the number of genes duplicated in each *Xenarthran* lineage inferred by maximum likelihood. Branch lengths are drawn proportional to the per gene duplication rate (see inset scale) and colored according to lineage relationships shown in *Figure 1*. (**B**) Word cloud of Reactome pathways in which *Xenarthan* (green) and *Pilosan* (dark green) gene duplicates are enriched (FDR q≤0.25). Pathway term sizes are scaled according to the −log₁₀ hypergeometric p-value of their enrichment (see inset scale).

The online version of this article includes the following source data for figure 3:

**Source data 1.** Genes duplicated in the stem-lineage of *Xenarthra*.

**Source data 2.** Genes duplicated in the stem-lineage of *Pilosa*.

**Source data 3.** Genes duplicated in the stem-lineage of *Cingulata*.

**Source data 4.** Pathways in which genes duplicated in the stem-lineage of *Xenarthra* are enriched.

**Source data 5.** Pathways in which genes duplicated in the stem-lineage of *Pilosa* are enriched.

**Source data 6.** Pathways in which genes duplicated in the stem-lineage of *Cingulata* are enriched.

jumps in traits and has previously been shown to out-perform other models of body mass evolution, including standard Brownian motion models, Ornstein–Uhlenbeck models, early burst maximum like-lihood models, and heterogeneous multi-rate models (*Elliot and Mooers, 2014*). Finally, we used extant and reconstructed body mass and lifespan data to estimate intrinsic cancer risk ($K$) via a simpli-fied multistage cancer risk model: $K \approx Dt^6$, where $D$ is maximum body size and $t$ is the maximum lifespan across *Xenarthran* lineages. As expected, we found that body mass and lifespan data were correlated, that large-bodied lineages tended to also be long-lived, and that intrinsic cancer risk in *Xenarthra* also varies with changes in body size and longevity, with particularly notable decreases in intrinsic cancer risk in the *Xenarthran* stem-lineage, giant armadillo, Glyptodont, giant anteater, Mylodon, and Giant ground sloth (*Figure 2*).

## Pervasive duplication of tumor suppressors in *Xenarthra* and *Pilosa*

We previously found that reductions in intrinsic cancer risk in *Afrotherians*, particularly in the stem-lineage of *Proboscideans*, was associated with pervasive duplication of genes in tumor suppressor pathways (*Vazquez and Lynch, 2021*). To test if duplication of tumor suppressors was similarly common in *Xenarthran* lineages with reduced cancer risk, we first identified duplicated genes in the genomes of *Afrotherians* (*Orycteropus afer*, *Trichechus manatus*, *Procavia capensis*, and *Loxodonta africana*) and *Xenarthrans*, including three sloths (*Bradypus variegatus*, *Choloepus didactylus*, and *Choloepus hoff-manni*), three armadillos (*Chaetophractus vellerosus*, *D. novemcinctus*, and *Tolypeutes matacus*), and two anteaters (*Myrmecophaga tridactyla* and *Tamandua tetradactyla*). Next, we inferred the lineage(s) in which each duplication occurred using maximum likelihood-based ASR implemented in IQ-TREE2 (*Minh et al., 2020*), and tested if duplicate genes were enriched in Reactome pathways (*Jassal et al., 2020*) using the hypergeometric test implemented in WebGestalt (*Liao et al., 2019*). We thus identi-fied 149 duplicates in the *Xenarthran* stem-lineage, 139 duplicates in the Cingulate stem-lineage, and 147 duplicates in the *Pilosan* stem-lineage; very few duplicate genes were identified in other *Xenar-thran* lineages (*Figure 3A*). Duplicates in the *Xenarthran* stem-lineage were enriched in 92 pathways (FDR q-value≤0.25), of which 26 (28%) were related to cancer biology such as cell cycle regulation, protein folding, intrinsic apoptosis, and regulation of p53 degradation, while duplicates in the *Pilosan*

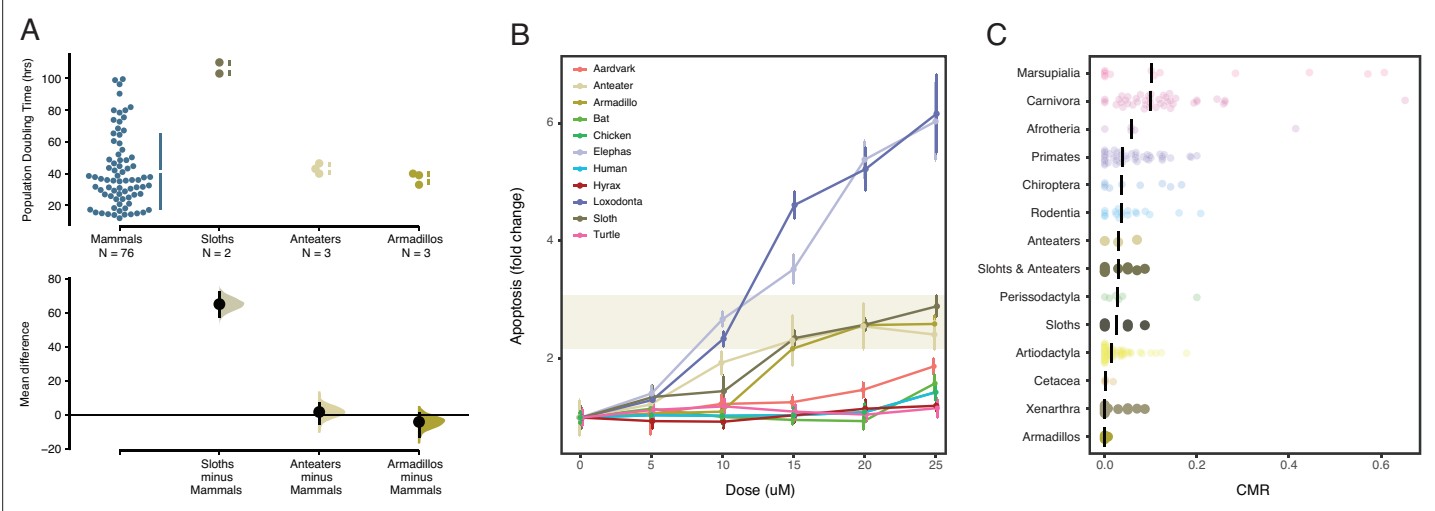

**Figure 4.** *Xenarthrans* have anticancer cellular phenotypes and a low prevalence of cancer. (**A**) Population doubling time estimates for mammalian cells (upper) and mean differences (lower). Mean differences are depicted as a dot with 95% confidence intervals as vertical error bars; p values are reported from a two-sided permutation t-test with 5000 reshuffles of control and test labels. The unpaired mean difference between mammals and sloths is 65.1 (57.5–71.8), p=4.0×10⁻⁴. The unpaired mean difference between mammals and anteaters is 1.71 (5.0–7.16), p=0.905. The unpaired mean difference between mammals and armadillos is –4.1 (12.5–0.86), p=0.781. (**B**) Induction in apoptosis in vertebrate cells by mitomycin C. Data are shown as fold change in caspase 3/7 activity (apoptosis), mean (dot) and standard deviation (vertical line); n=12 for each data point. (**C**) Cancer prevalence data for each species (dot) grouped by clade and shown as jittered stripchart. The number of species per clade is: Afrotheria=5, Artiodactyla=68, Carnivora=43, Cetacea=4, Chiroptera=9, Marsupialia=13, Perissodactyla=6, Primates=47, Rodentia=18, anteaters=3, sloths=6, armadillos=5, and Xenarthra=14. Medians for each clade are shown with vertical black bars.

The online version of this article includes the following source data for figure 4:

**Source data 1.** Data plotted in *Figure 4A*.

**Source data 2.** Data plotted in *Figure 4B*.

**Source data 3.** Data plotted in *Figure 4C*.

stem-lineage were enriched in 13 pathways (FDR q-value≤0.25), of which 8 (61%) were related to cancer biology pathways involving the cell cycle (*Figure 3B*). In contrast, duplicates in the Cingulate stem-lineage were only enriched in a single pathway (FDR q-value≤0.25), 'Cholesterol biosynthesis' (enrichment ratio=15.87, hypergeometric p=1.00×10⁻⁴, FDR q=0.18) which unlike the *Xenarthran* and *Pilosan* duplicates appear to be unrelated to cancer biology. These observations are consistent with earlier studies suggesting that *Xenarthrans* have extra copies of tumor suppressor genes (*Tollis et al., 2020*; *Vazquez and Lynch, 2021*).

## Sloth cells have extremely long doubling times

The enrichment of gene *Xenarthran* and Pisolan duplications in pathways related to the cell cycle suggest that cells from these species may have different cell cycle dynamics than other species. While direct measurements of cell cycle phase length or overall duration are not available for most species, many studies report population doubling times from mammalian cells in culture; population doubling time is proportional to the cumulative length of individual cell cycle phases, particularly the length of $G_1$ (*Blank et al., 2018*). Therefore we performed a thorough literature review and assembled a data set of population doubling times from 76 species representing all mammalian orders (*Figure 4—source data 1*) and directly estimated population doubling time for primary fibroblasts of eight *Xenarthan* species: three species of armadillo (Southern three-banded armadillo, *T. matacus*; screaming hairy armadillo, *C. vellerosus*; and six-banded armadillo, *Euphractus sexcinctus*); two species of anteater (Northern tamandua, *Tamandua mexicana*; Southern tamandua, *T. tetradactyla tetradactyla*); and two species of sloth (Linnaeus's two-toed sloth, *C. didactylus*; Hoffmann's two-toed sloth, *C. hoffmanni*).

While there was considerable variation in population doubling times between mammalian cells, armadillo (unpaired mean difference=–4.1, two-sided permutation t-test p-value=0.09) and anteater (unpaired mean difference=1.7, two-sided permutation t-test p-value=0.78) cells had similar doubling

times as other mammals (*Figure 4A*). Cells of both sloth species, however, had significantly longer doubling times than cell lines from other all other species with an unpaired mean difference in doubling time of 65.1 hr (95% confidence interval [CI]: 57.5–71.8) and a two-sided permutation t-test p-value=4.0×10⁻⁴ (*Figure 4A*). Our doubling time estimates for Hoffmann's two-toed sloth (103 hr) is similar to a previously published estimate for this species (107 hr), and our doubling time estimates for Northern tamandua (43 hr) and Southern tamandua (40 hr) are similar to a previously published estimate (46 hr) for the giant anteater (*M. tridactyla*) suggesting these doubling times are reliable (*Gomes et al., 2011*). Our results suggest that sloth cells have the longest reported doubling time for any mammal, which may allow for more time to correct DNA damage and other genomic instabilities such as aneuploidy. However, we note that doubling times may reflect biological differences in cell cycle length between species as well as technical differences that are not directly related to species-specific differences in doubling time or cell cycle length.

## *Xenarthran* cells are very sensitive to DNA damage

Our observation that duplicate genes in the *Xenarthran* stem-lineage are enriched in pathways related to apoptosis, DNA damage responses, and the p53 signaling pathway such as 'Intrinsic Pathway for Apoptosis' (enrichment ratio=6.76, hypergeometric p=9.75×10⁻³, FDR q=0.20), 'Inhibition of replication initiation of damaged DNA by RB1/E2F1' (enrichment ratio=15.26, hypergeometric p=7.30×10⁻³, FDR q=0.17), 'Transcriptional Regulation by TP53' (enrichment ratio=2.76, hypergeometric p=3.22×10⁻³, FDR q=0.11), and 'Regulation of TP53 Degradation' (enrichment ratio=8.50, hypergeometric p=5.14×10⁻³, FDR q=0.14) suggests that *Xenarthran* cells may respond to genotoxic stresses differently than other species. Indeed, we previously compared the sensitivity of African elephant, Asian elephant, South African rock hyrax, East African aardvark, and Southern three-banded armadillo primary dermal fibroblasts to DNA damage induced with either mitomycin C, doxorubicin, or UV-C and found that while elephant cells had the greatest apoptotic response, Southern three-banded armadillo cells had a greater apoptotic response than aardvark and hyrax cells but less than elephant cells (*Sulak et al., 2016*). As this previous study was focused on the DNA damage response in elephant cells, it only included a single *Xenarthran* species, and so we could not determine if the behavior of Southern three-banded armadillo cells was typical for *Xenarthrans*.

To explore this observation in greater detail, we repeated our assay of apoptosis in response to mitomycin C using primary dermal fibroblasts collected from African and Asian elephants, South African rock hyrax, East African aardvark, Southern three-banded armadillo, Southern tamandua, Hoffmann's two-toed sloth, human, little brown bat, chicken, and turtle. Mitomycin C (MMC) was selected as it induces multiple types of DNA damage (*Lee et al., 2006*), and we multiplexed measurements of apoptosis, necrosis, and cell cycle arrest using the ApoTox-Glo assay (*Figure 4B*). Consistent with our previous study (*Sulak et al., 2016*), we found that Southern three-banded armadillo, Southern tamandua, and Hoffmann's two-toed sloth cells induced apoptosis at lower MMC doses and had a greater response effect size than South African rock hyrax, East African aardvark, human, bat, chicken and turtle cells, but less than African and Asian elephant cells. These data indicate that *Xenarthran* fibroblasts are particularly sensitive to DNA damage.

## Extremely low cancer prevalence in *Xenarthrans*, especially armadillos

Previous studies have reported sporadic cases of cancer in sloths (*do Amaral et al., 2022*; *Linnehan et al., 2019*; *Salas et al., 2014*) and anteaters (*Madsen et al., 2017*; *Sanches et al., 2013*) but there have been few systematic studies of cancer prevalence in *Xenarthra*. Remarkably, *Vincze et al., 2022* found only one case of neoplasia in 22 necropsies of Linnaeus's two-toed sloth (*C. didactylus*) suggesting that cancer is uncommon in at least one sloth species. To explore cancer prevalence in *Xenarthra* in greater detail, we gathered published mortality data from *Pilosa*, including giant anteater (*M. tridactyla*), Southern tamandua (*T. tetradactyla*), silky anteater (*Cyclopes* sp.), maned three-toed sloths (*Bradypus torquatus*), brown-throated sloths (*B. variegatus*), pale-throated sloth (*Bradypus tridactylus*), Linné's two-toed sloth (*C. didactylus*), as well as from a pooled set of three-toed sloths (*Bradypus* sp.) and two-toed slots (*Choloepus* sp.), from four published surveys (*Arenales et al., 2020a*; *Arenales et al., 2020b*; *Diniz and Oliveira, 1999*; *Diniz et al., 1995*). These data indicate that neoplasia prevalence was only 1.8% (95% CI: 0.02–6.5%) in sloths; 2.3% (95% CI: 0.08–5.7%) in anteaters; and 2.2% (95% CI: 0.09–4.6%) in *Pilosa* (*Table 1*, *Figure 4C*).

**Table 1.** Cancer prevalence in *Xenarthrans*.

Clopper-Pearson exact 95% CIs are reported for each taxa, asterisks (*) indicate taxa with more than 82 pathology reports. Data from more than two studies are indicated with 'combined'.

| Common name | Species name | Necropsies | Neoplasia | Prevalence (95% CI) |
|---|---|---|---|---|
| Southern tamandua (combined) | *Tamandua tetradactyla* | 60 | 1 | 0.07 (0.0004–0.089) |
| Giant anteater (combined)* | *Myrmecophaga tridactyla* | 139 | 4 | 0.029 (0.008–0.072) |
| Silky anteater | *Cyclopes* sp. | 3 | 0 | 0 (0–0.708) |
| Maned three-toed sloth | *Bradypus torquatus* | 25 | 0 | 0 (0–0.137) |
| Pale-throated sloth | *Bradypus tridactylus* | 2 | 0 | 0 (0–0.841) |
| Brown-throated sloth | *Bradypus variegatus* | 8 | 0 | 0 (0–0.369) |
| Three-toed sloths (combined) | *Bradypus* sp. | 69 | 0 | 0 (0–0.050) |
| Linné's two-toed sloth (combined) | *Choloepus didactylus* | 23 | 2 | 0.087 (0.011–0.28) |
| Two-toed sloths (combined) | *Choloepus* sp. | 40 | 2 | 0.050 (0.006–0.169) |
| Nine-banded armadillo (combined)* | *Dasypus novemcinctus* | 275 | 2 | 0.007 (0.0009–0.026) |
| Six-banded armadillo | *Euphractus sexcinctus* | 48 | 0 | 0 (0–0.074) |
| Naked-tailed armadillos | *Cabassous* sp. | 5 | 0 | 0 (0–0.522) |
| Three-banded armadillo | *Tolypeutes* sp. | 4 | 0 | 0 (0–0.602) |
| Giant armadillo | *Priodontes maximus* | 1 | 0 | 0 (0–0.975) |
| Vermilingua (anteaters)* | | 202 | 5 | 0.023 (0.008–0.057) |
| Folivora (sloths)* | | 109 | 2 | 0.018 (0.002–0.065) |
| Pilosa (sloths and anteaters)* | | 311 | 7 | 0.022 (0.009–0.046) |
| Cingulata (armadillos)* | | 333 | 2 | 0.006 (0.0007–0.021) |
| Xenarthra (sloths, anteaters, armadillos)* | | 644 | 9 | 0.014 (0.006–0.026) |

The online version of this article includes the following source data for table 1:

**Source data 1.** Vital statistics for armadillos in the NHDP colony 2010–2020.

Similarly, previous studies have reported individual cases of cancer in nine-banded armadillos (*Lee et al., 2015*; *Madsen et al., 2017*; *Pence et al., 1983*), including a thalidomide-induced malignant choriocarcinoma that perforated the uterus and metastasized to the liver, spleen, mesentery, and lungs (*Marin-Padilla and Benirschke, 1963*). In the only study to systematically evaluate cancer prevalence, *Boddy et al., 2020* found only two cases of neoplasia (leiomyoma and bronchial adenoma) among 69 necropsies of nine-banded armadillos (*D. novemcinctus*), suggesting cancer is rare in armadillos. Therefore, we explored neoplasia prevalence in armadillos from published mortality data and found that no neoplasias were reported among 55 nine-banded armadillos, 48 six-banded armadillos (*E. sexcinctus*), 5 greater naked-tailed armadillos (*Cabassous* sp.), 4 three-banded armadillos (*Tolypeutes* sp.), and 1 giant armadillo (*Priodontes maximus*) (*Diniz et al., 1997*). This remarkably low neoplasia prevalence in armadillos prompted us to compile neoplasia prevalence in a large colony of wild-caught nine-banded armadillos with extensive post-mortem veterinary pathology reports managed by the National Hansen's Disease Program (NHDP) Laboratories. We surveyed 153 pathology reports from *D. novemcinctus* from 2010 to 2020 and found no cases of neoplasia. When combined with neoplasia prevalence data from *Boddy et al., 2020* and *Diniz et al., 1997*, the estimated cancer prevalence in *D. novemcinctus* was only 0.73% (95% CI: 0.09–2.6%) in 275 individuals (*Table 1*, *Figure 4C*).

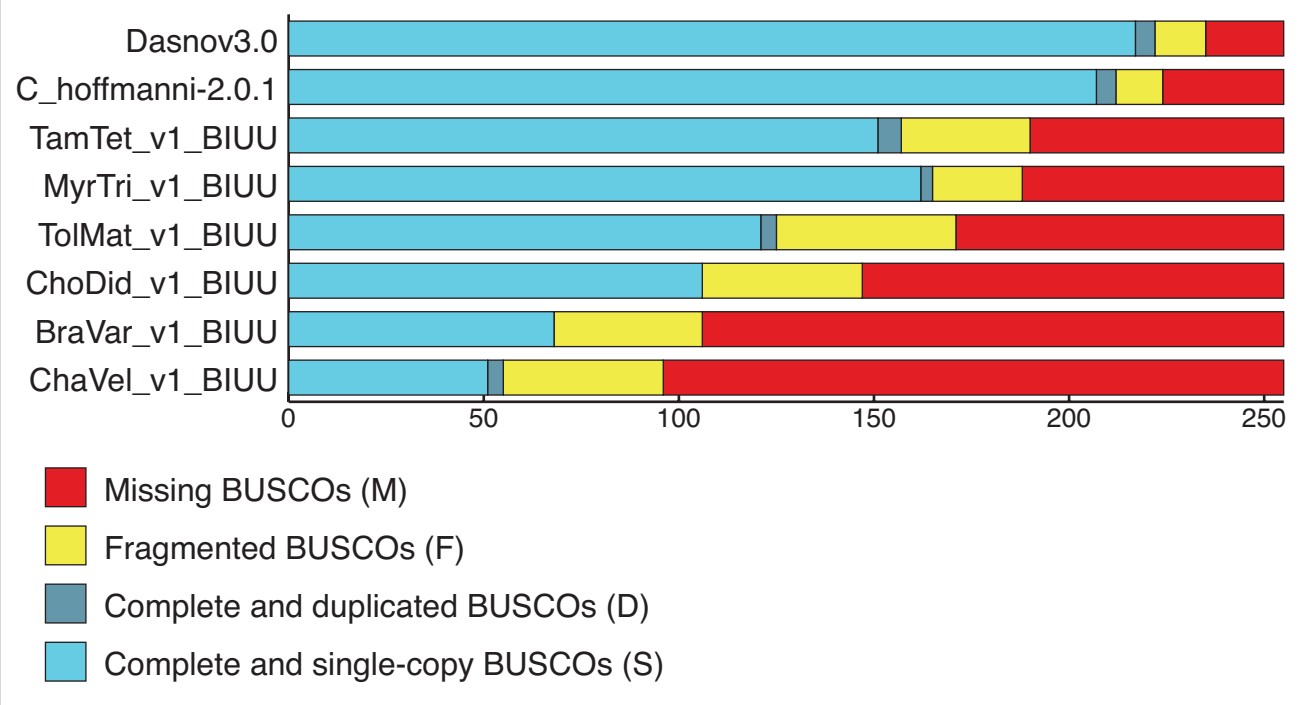

**Figure 5.** BUSCO scores for *Xenarthran* genomes used to estimate gene copy number. BUSCO results from Galaxy (version 4.1.4) using the *Eukaryota* lineage. The number of complete and single copy (light blue), complete and duplicated (dark blue), fragmented (yellow), and missing (red) genes are shown. We note that the number of duplicated genes identified by BUSCO is inversely proportional to the number of BUSCO genes missing in each genome, suggesting duplicate gene counts in our data may be underrepresented.

Next, we compared neoplasia prevalence in *Xenarthra* to other *Therian* mammals from two published studies that included pathology reports from 37 (*Boddy et al., 2020*) and 191 (*Vincze et al., 2022*) species of *Therian* mammals, as well as three species of cetaceans (*Albuquerque et al., 2018*). The total data set includes cancer prevalence data from 221 species. As a group, the mean prevalence of neoplasia in *Xenarthra* was 1.4% (95% CI: 0.6–2.7%), in contrast the mean prevalence of neoplasia in across other *Eutherian* mammals was 17.57% (95% CI: 14.86–20.53%), but there was much variation across species and orders (*Figure 4C*). Cancer prevalence was lowest among armadillos (*Cingulata*) and *Xenarthra* (*Figure 4C*). Thus, we conclude that *Xenarthrans*, and especially armadillos, are particularly cancer-resistant mammals.

## Caveats and limitations

This study has several inherent limitations. First, genome quality can play an important role in our ability to identify duplicate genes and most *Xenarthran* species lack high-quality reference genomes (*Figure 5*) or transcriptome annotations making the assessment of functional gene duplication difficult for some species (*Vazquez and Lynch, 2021*). We observed that the number of duplicated genes identified by BUSCO, however, is inversely proportional to the number of BUSCO genes missing in each genome, which suggests that the quality of these genomes may underrepresent, not overrepresent, the number of identified gene duplications in our study. We also assume that gene duplicates maintain ancestral tumor suppressor functions and increase cancer resistance through either dosage effects, redundancy to loss of function mutations, or sub- and neofunctionalization that conserve their tumor suppressor properties. However, sub- and neofunctionalization between paralogs can also lead to functional divergence (*Innan and Kondrashov, 2010*), and developmental systems drift (*Lynch, 2009*; *True and Haag, 2001*) can lead to functional divergence between orthologs and incorrect pathway assignment (*Altenhoff et al., 2012*; *Liao and Zhang, 2008*; *Stamboulian et al., 2020*).

Second, while we have shown that fibroblasts from *Xenarthra* are particularly sensitive to DNA damage, and that sloths cells have very long doubling times, these studies used only a single- cell type from a single individual despite likely variation in these traits across cell types and individuals. We

have also not shown that these duplicate genes functionally contribute to cancer resistance phenotypes. However, our observation that cells from different vertebrate classes and mammalian orders have similar apoptotic responses to MMC; and our replication of previously published doubling times for some species suggests that these species effects are greater than possible effects from individual variation.

The estimation of cancer prevalence can be particularly error-prone for species without long-term morbidity and mortality studies. The probability of detecting at least one individual with cancer, for example, is strongly dependent on sample size. Indeed, cancer was detected in at least one individual in almost all species with more than 82 individual pathological records, illustrating that cancer is likely to be detected in all mammals with adequate sample sizes (*Vincze et al., 2022*). The number of post-mortem pathology reports we gathered from the literature for *Xenarthrans* was below 82 for all species except giant anteater and nine-banded armadillo suggesting caution when interpreting the low cancer prevalence for other species. However, our data set included 275 pathology reports from *D. novemcinctus*, of which only two had reports of neoplasia and no reports of natural metastasis, indicating that the low prevalence of cancer in *D. novemcinctus* is reliable. Finally, while cancer prevalence from 333 armadillos is extremely low at 0.6% (95% CI: 0.07–2.1%), our data set only includes armadillo species from the family *Dasypodidae* and additional data from the other armadillo family (*Chlamyphoridae*) is necessary to determine if this low cancer prevalence is a common trait of armadillos (*Cingulata*) or limited to *Dasypodidae*.

## Ideas and speculation

Extant *Xenarthrans* are relatively small bodied: for example, sloths generally weigh 3.5–5.5 kg, armadillos weigh 120 g to 33 kg, and tamanduas and anteaters weigh 400 g to 28.5 kg. In stark contrast, recently extinct sloth and armadillo species that lived during the *Pleistocene* were much larger: the extinct armadillo *Glyptodon* is estimated to have been roughly the size of a Volkswagen Beetle, weighing 800–840 kg; and the ground sloth *Megatherium americanum* is one of the largest known land mammals weighing nearly 4000 kg and measuring up to 6 m in length (*Raj Pant et al., 2014*). Remarkably, a previous study found that the rate of body size evolution in *Megatheriidae* was very rapidly, perhaps as fast as 129 kg/My (95% CI: 42–384 kg/My), while the ~5.5 kg *Choloepus* was deeply nested within a clade with an average body mass of 236 kg (*Raj Pant et al., 2014*). These data indicate that the living *Xenarthrans* represent a biased sample of *Xenarthran* body sizes throughout history, particularly for sloths and armadillos, and have the developmental potential to rapidly evolve large bodies. Thus, they must also have evolved mechanisms that reduce their cancer risk.

Our data suggest that a burst of tumor suppressor duplication in the *Xenarthran* and *Pilosan* stem-lineages may have contributed to the evolution of this reduced cancer risk, at least in part by increasing the sensitivity of *Xenarthran* cells to DNA damaging agents, which prevents the accumulation of mutations that give rise to cancer as a byproduct of DNA repair. Perhaps surprisingly, we found that this burst of tumor suppressor gene duplication occurred *prior to* the evolution of large body sizes in *Xenarthrans*, suggesting that the evolution of reduced cancer risk was most likely not a response to selection for increased body size, but rather may have allowed it. We similarly found that duplication of genes with tumor suppressor functions was relatively common in *Afrotherians* prior to the independent evolution of large bodies sizes. For example, a burst of tumor suppressor duplication occurred in the ancestor of *Paenungulates*, which was relatively small, before the evolution of large-bodied species such as the extinct *Titanohyrax*, which is estimated to have weighed~1300kg (*Schwartz et al., 1995*), the extinct Stellar's sea cow which is estimated to have weighed~8000–10,000kg (*Scheffer, 1972*), and gigantic extinct *Proboscideans* such as *Deinotherium* (~12,000kg), *Mammut borsoni* (16,000kg), and the straight-tusked elephant (~14,000kg) (*Larramendi, 2015*). Therefore an abundance of tumor suppressor genes and/or reduced cancer risk may be an exaptation or epistatic facilitator for the evolution of large bodies rather than a directly selected effect of increased body size.

## Conclusions

While extant *Xenarthrans* are small bodied, recently extinct *Xenarthrans* such as *Megatherium* were among the largest known land mammals; their large body size implies that they must have had exceptional cancer suppression mechanisms in order to resolve Peto's Paradox. We found that intrinsic cancer risk was dramatically reduced in the stem-lineages of *Xenarthra*, *Pilosa*, and *Cingulata*, as well

as in several extinct large *Xenarthran* species; and that these changes coincided with bursts of duplication of tumor suppressor genes. These genes were enriched in pathways that regulate apoptosis and the cell cycle which may be related to the evolution of anti-cancer cellular phenotypes in the *Xenarthran* lineage. We found that sloth, armadillo, and anteater cells, for example, were very sensitive to DNA damage induced by the genotoxic drug mitomycin C. Repair of mitomycin C induced DNA damage involves multiple pathways such as nucleotide excision repair, homologous recombination repair, and translesion bypass pathways (*Lee et al., 2006*), suggesting these pathways have evolved *Xenarthran*-specific changes leading to their increased sensitivity. We additionally show that pathways related to the cell cycle are enriched among genes that duplicated in the stem-lineage of *Xenarthra* and *Pilosa*, and that cell cycle dynamics (as measured by doubling times of cells in culture) are longer in sloths compared to other mammals. Finally, our data suggest that armadillos, which are already studied in the biomedical community because they are the sole animal model of leprosy (*Adams et al., 2012*; *Kirchheimer and Storrs, 1971*; *Storrs et al., 1975*), have particularly low cancer prevalence and apparent lack of natural metastatic cancer despite the ability to experimentally induce metastasis (*Marin-Padilla and Benirschke, 1963*). Taken together, our study suggests that future studies on Peto's Paradox should not only focus on extant large-body organism, but also on clades of ancestrally large-bodies species such as *Xenarthra*; and that armadillos may be a promising model organism to study the mechanisms that underlie cancer initiation and progression.

## Materials and methods
### Ancestral body size reconstruction

We first assembled a time-calibrated supertree of *Eutherian* mammals by combining the time-calibrated molecular phylogeny of *Bininda-Emonds et al., 2007*, the time-calibrated total evidence *Afrotherian* phylogeny from *Puttick and Thomas, 2015* and the time-calibrated *Xenarthran* phylogeny of *Delsuc et al., 2016*. To construct this *Eutherian* supertree, we replaced the *Afrotherian* and *Xenarthran* clades in the *Bininda-Emonds et al., 2007* phylogeny with the phylogenies of *Puttick and Thomas, 2015* and *Delsuc et al., 2016*, respectively, using Mesquite. Next, we jointly estimated rates of body mass evolution and reconstructed ancestral states using a generalization of the Brownian motion model that relaxes assumptions of neutrality and gradualism by considering increments to evolving characters to be drawn from a heavy-tailed stable distribution (the 'Stable Model') implemented in StableTraits (*Elliot and Mooers, 2014*). The stable model allows for large jumps in traits and has previously been shown to out-perform other models of body mass evolution, including standard Brownian motion models, Ornstein–Uhlenbeck models, early burst maximum likelihood models, and heterogeneous multi-rate models (*Elliot and Mooers, 2014*).

### Estimating the evolution of cancer risk

The dramatic increase in body mass and lifespan in some *Afrotherian* lineages, and the relatively constant rate of cancer across species of diverse body sizes, indicates that those lineages must have also evolved reduced cancer risk. To infer the magnitude of these reductions we estimated differences in intrinsic cancer risk across extant and ancestral *Afrotherians*. Following *Peto, 2015*, we estimate the intrinsic cancer risk ($K$) as the product of risk associated with body mass and lifespan. To determine ($K$) across species and at ancestral nodes (see below), we first estimated ancestral lifespans at each node. We used Phylogenetic Generalized Least-Square Regression (PGLS), using a Brownian covariance matrix as implemented in the R package *ape* (*Paradis et al., 2004*), to calculate estimated ancestral lifespans using our estimates for body size at each node. To estimate the intrinsic cancer risk of a species, we first inferred lifespans at ancestral nodes using PGLS and the model:

$$ln(lifespan) = \beta_1 corBrowninan + \beta_2 ln(size) + \epsilon \tag{1}$$

Following our method in *Vazquez and Lynch, 2021*, we calculate the intrinsic cancer risk $K$ at all nodes, using the simplified multistage cancer risk model described in *Peto, 2015* for body size and $D$ and lifespan $t$: $K \approx Dt^6$ . The fold change in cancer risk between a node and its ancestor was then defined as $log_2 \left( \frac{K_2}{K_1} \right)$ , where $K_1$ represents the cancer risk of the given node, and $K_2$ represents the cancer risk of its ancestral node.

## BUSCO assessment of genome assembly and annotation completeness

We used BUSCO (*Manni et al., 2021*) as implemented on Galaxy (version 4.1.4) to assess genome assembly and annotation completeness of publicly available *Xenarthran* genomes, including nine-banded armadillo (*D. novemcinctus*: dasNov3), Screaming hairy armadillo (*C. vellerosus*: ChaVel_v1_BIUU), Southern three-banded armadillo (*T. matacus*: TolMat_v1_BIUU), Hoffman's two-toed sloth (*C. hoffmannii*: C_hoffmannii-2.0.1_DNAZoo_HiC), Linnaeus's two-toed sloth (*C. didactylus*: ChoDid_v1_BIUU), Brown-throated three-toed sloth (*B. variegatus*: BraVar_v1_BIUU), Southern tamandua (*T. tetradactyla*: TamTet_v1_BIUU), and giant anteater (*M. tridactyla*: MyrTri_v1_BIUU). BUSCO was run using the settings 'Mode=Genome and Lineage=Eukaryota'. We assessed genome quality and completeness using core *Eukaryota* genes because this gene set is the most conserved and thus provides a robust estimation of completeness to guide our methods to identify duplicate genes (*Figure 5*).

## Identification of gene duplications using reciprocal best-hit BLAT and estimated copy number by coverage

We previously developed a reciprocal best-hit BLAT (RBHB) pipeline to identify putative homologs and estimate gene copy number across species (*Vazquez and Lynch, 2021*). The RBHB strategy to identify duplicate genes is conceptually straightforward: (1) Given a gene of interest $G_A$ in a query genome $A$, one searches a target genome $B$ for all possible matches to $G_A$ ; (2) For each of these hits, one then performs the reciprocal search in the original query genome to identify the highest-scoring hit; (3) A hit in genome $B$ is defined as a homolog of gene $G_A$ if and only if the original gene $G_A$ is the top reciprocal search hit in genome $A$. We selected BLAT as our algorithm of choice, as this algorithm is sensitive to highly similar (>90% identity) sequences, thus identifying the highest-confidence homologs while minimizing many-to-one mapping problems when searching for multiple genes. RBHB performs similar to other more complex methods of orthology prediction and is particularly good at identifying incomplete genes that may be fragmented in low-quality/poorly assembled regions of the genome (*Hernández-Salmerón and Moreno-Hagelsieb, 2020*; *Johnson, 2007*; *Kent, 2002*; *Vazquez and Lynch, 2021*; *Ward and Moreno-Hagelsieb, 2014*).

In fragmented genomes, many genes will be split across multiple scaffolds, which results in BLA(S)T-like methods calling multiple hits when in reality there is only one gene. To compensate for this, we utilized the statistic estimated copy number by coverage (ECNC), which averages the number of times we hit each nucleotide of a query sequence in a target genome over the total number of nucleotides of the query sequence found overall in each target genome (*Vazquez and Lynch, 2021*). This allows us to correct for genes that have been fragmented across incomplete genomes, while accounting for missing sequences from the human query in the target genome. Mathematically, this can be written as:

$$ECNC = \frac{\sum_{n=1}^{l} C_n}{\sum_{n=1}^{l} bool\left(C_n\right)} \tag{2}$$

where $n$ is a given nucleotide in the query, $l$ is the total length of the query, $C_n$ is the number of instances that $n$ is present within a reciprocal best-hit, and $bool\left(C_n\right)$ is 1 if $C_n > 0$ or 0 if $C_n = 0$.

## RecSearch pipeline and query gene set

We previously described a Python pipeline for automating our RBHB/ECNC search strategy between a single reference genome and multiple target genomes using a list of query sequences from the reference genome (*Vazquez and Lynch, 2021*), and utilized the same methods for this study. Briefly, for the query sequences in our search, we used the hg38 UniProt proteome (*The UniProt Consortium, 2017* ), which is a comprehensive set of protein sequences curated from a combination of predicted and validated protein sequences generated by the UniProt Consortium. Next, we excluded genes from downstream analyses for which assignment of homology was uncertain, including uncharacterized ORFs (991 genes), LOC (63 genes), HLA genes (402 genes), replication-dependent histones (72 genes), odorant receptors (499 genes), ribosomal proteins (410 genes), zinc finger transcription factors (1983 genes), viral and repetitive-element-associated proteins (82 genes) 'Uncharacterized', 'Putative', or 'Fragment' proteins (30,724 genes), leaving a final set of 37,582 query protein isoforms, corresponding to 18,011 genes. We then searched for all copies of 18,011 query genes in publicly

available *Xenarthran* genomes using the RBBH/ECNC approach described above (*Vazquez and Lynch, 2021*).

## Validation of gene duplications

Our assessments of genome quality using BUSCO indicate that highest quality genomes were nine-banded armadillo (*D. novemcinctus*: dasNov3) and Hoffman's two-toed sloth (*C. hoffmannii*: C_hoffmannii-2.0.1_DNAZoo_HiC), while the other genomes were of much lower quality as assessed by the large number of missing BUSCOS which was 20 and 31 for nine-banded armadillo (*D. novemcinctus*: dasNov3) and Hoffman's two-toed sloth (*C. hoffmannii*: C_hoffmannii-2.0.1_DNAZoo_HiC), respectively, and ranged from 65 to 159 for the other genomes (*Figure 5*). These data suggest that computational inferences of copy number variation will lead to many incorrect inferences of gene duplication and loss in the lower quality genomes. Therefore, we cross-referenced the duplicates identified by our RBHB/ECNC pipeline with Ensembl inferred same-species paralogies (within_species_paralog) to identify a 'high-quality' set of duplicated genes (*Altenhoff and Dessimoz, 2009*), that is, genes inferred as duplicated via multiple methods.

Next, we used Ensembl BioMart to download the cDNA sequences for the 'high-quality' duplicated genes from nine-banded armadillo (*D. novemcinctus*: dasNov3) and Hoffman's two-toed sloth (*C. hoffmannii*: choHof1). As public RNA-seq data were not available for validating duplications in other species, we used BLAST (discontiguous megablast) to manually determine the copy number of these genes in the lower quality Screaming hairy armadillo (*C. vellerosus*: ChaVel_v1_BIUU), Southern three-banded armadillo (*T. matacus*: TolMat_v1_BIUU), Hoffman's two-toed sloth (*C. hoffmannii*: C_hoffmannii-2.0.1_DNAZoo_HiC), Linnaeus's two-toed sloth (*C. didactylus*: ChoDid_v1_BIUU), Brown-throated three-toed sloth (*B. variegatus*: BraVar_v1_BIUU), Southern tamandua (*T. tetradactyla*: TamTet_v1_BIUU), and giant anteater (*M. tridactyla*: MyrTri_v1_BIUU) genomes. Note that this approach allows us to determine if 'high-quality' duplicated genes in the nine-banded armadillo (*D. novemcinctus*: dasNov3) and Hoffman's two-toed sloth (*C. hoffmannii*: choHof1) are present (ancestral) in *Xenarthran* lineages but cannot identify gene duplications unique (derived) in those lineages.

## Reconstruction of ancestral copy numbers

We encoded the copy number of each gene for each species as a discrete trait ranging as 0 (one gene copy) or 1 (duplicated) and used IQ-TREE2 (*Minh et al., 2020*) and ModelFinder (*Kalyaanamoorthy et al., 2017*) to select the best-fitting model of character evolution, which was inferred to be a General Time Reversible type model for morphological data (GTR2) with empirical character state frequencies (F0). Gene duplication events across the phylogeny were identified with the empirical Bayesian ASR method implemented in IQ-TREE2 (*Minh et al., 2020*), the best fitting model of character evolution (GTR2+F0), and the unrooted species tree for *Atlantogenata*. We considered ASRs to be reliable if they had Bayesian Posterior Probability (BPP) ≥0.80; less reliable reconstructions were excluded from pathway analyses. Note that in *Vazquez and Lynch, 2021*, copy number states were coded as multi-state data corresponding to the actual copy number of each duplicate. However, since many *Xenarthran* genomes are relatively low quality, the exact number of duplicate genes in each genome is unreliable for some species. Thus, to account for the fact, we used binary coding for gene copy number.

## Pathway enrichment analysis

To determine if gene duplications were enriched in particular biological pathways, we used the WEB-based Gene SeT AnaLysis Toolkit (WebGestalt) to perform Over-Representation Analysis (ORA) using the Reactome database (*Jassal et al., 2020*; *Liao et al., 2019*). Gene duplicates in each lineage were used as the foreground gene set, and the initial query set was used as the background gene set. WebGestalt uses a hypergeometric test for statistical significance of pathway overrepresentation, which we refined using the Benjamini-Hochberg FDR multiple-testing correction in WebGestalt.

## Cell culture, doubling time estimation, and ApoTox-Glo assay

Primary dermal fibroblasts from African elephant (*L. africana*), Asian elephant (*Elephas maximus*), South African rock hyrax (*P. capensis*), East African aardvark (*O. afer*), Southern three-banded armadillo (*Tolypeutus matacus*; female, passage 6), screaming hairy armadillo (*C. vellerosus*; female, passage

6), six-banded armadillo (*E. sexcinctus*; female, passage 6), Northern tamandua (*T. mexicana*; male, passage 6), Southern tamandua (*T. tetradactyla tetradactyla*; female, passage 6), Linnaeus's two-toed sloth (*C. didactylus*; male, passage 6), Hoffmann's two-toed sloth (*C. hoffmanni*; male, passage 10), human, bat (*Myotis lucifugus*, passage 14), turtle (*Terrapene carolina*, passage 14), and chicken (*Gallus gallus*, passage unknown) were maintained in T-75 culture flasks in a humidified incubator chamber at 37°C with 5% $CO_2$ in a culture medium consisting of FGM/EMEM (1:1) supplemented with insulin, FGF, 10% FBS and Gentamicin/Amphotericin B (FGM-2, singlequots, Clonetics/Lonza). For regular passaging, cells were divided before reaching 80% confluency. For drug treatments, 10,000 cells were seeded into each well of an opaque bottomed 96-well plate, leaving a row with no cells (background control) and treated 12 hr after seeding with a serial dilution of mitomycin C (0, 5, 10, 15, 20, and 25 μM), and 12 biological replicates for each condition. After 12 hr of incubation, we measured cell viability, cytotoxicity, and caspase-3/7 activity using the ApoTox-Glo Triplex Assay (Promega) in a GloMax-Multi+Reader (Promega). Data were standardized to no-drug (0 μM) control cells. For doubling time estimation, cells were harvested from T75 flasks at 80% confluency and seeded into six-well culture plates at 10,000 cells/well. Cell counts were performed automatically using the confluence estimation method using a Cytation-5 multi-mode plate reader (BioTek), doubling time was estimated as the number of hours cells took to grow from ~20% to ~40% confluency. Cells were determined to be mycoplasma free, and the identity of the African elephant (*L. africana*), Asian elephant (*Elephas maximus*), South African rock hyrax (*P. capensis*), East African aardvark (*O. afer*), Southern three-banded armadillo (*Tolypeutus matacus*; female), screaming hairy armadillo (*C. vellerosus*), six-banded armadillo (*E. sexcinctus*; female), Northern tamandua (*T. mexicana*), Southern tamandua (*T. tetradactyla tetradactyla*), Linnaeus's two-toed sloth (*C. didactylus*), and Hoffmann's two-toed sloth (*C. hoffmanni*) cell lines confirmed by the San Diego Frozen Zoo (from which these cell lines were obtained).

## Mammalian population doubling time estimation

We gathered population doubling times from previously published studies (*Burkard et al., 2019*; *Burkard et al., 2015*; *Carvan et al., 1994*; *Wise et al., 2011*; *Wise et al., 2008*; *Li Chen et al., 2009*; *Rajput et al., 2018*; *Seluanov et al., 2008*; *Wang et al., 2011*; *Wang et al., 2021*; *Gomes et al., 2011*; *Annalaura et al., 2012*; *Yajing et al., 2018*). Some studies did not report doubling times but showed graphs of doubling times. For these data sets, doubling times were extracted from figures using WebPlotDigitizer Version 4.5 (*Rohatgi, 2021*).

## Cancer data collection from NHDP nine-banded armadillo

*Aktipis et al., 2015*, we defined any neoplastic growth as a cancer. The NHDP nine-banded armadillo (*D. novemcinctus*) colony is comprised of both wild-caught adult and captive-born sibling animals. The wild-caught adults, including pregnant females that subsequently gave birth to genetically identical quadruplets in captivity, were collected from various locations in Louisiana and Arkansas. Adult and juvenile animals were housed singly or in pairs in modified rabbit cages with soft plastic flooring that are ganged together with a tunnel to separate the sleeping and feeding area from litter pan area. The wild-caught adults received the following treatment upon entry into the colony: Penicillin G Procain (2 ml IM) repeated at 5 days, Praziquantel (0.4 ml IM at day 7), Ivermectin (0.1 ml in the food at day 14), and Prednisone 0.25 ml IM at day 21. The captive-born siblings were treated with Fenbendazole at ~6–10 weeks of age. At ~16 months of age, the animals were treated with Penicillin (1.0 ml) and dewormed with Ivermectin (0.1 ml) and Praziquantel (0.4 ml). Prednisone (10 mg/ml) was also given at this time. All animals were conditioned to captivity for ~1 year prior to experimentation.

To determine their immune response to *Mycobacterium leprae*, armadillos were injected intradermally with lepromin (heat-killed nude mice footpad derived *M. leprae* strain Thai 53). At 21 days, the sites were biopsied using a 4-mm punch biopsy and examined histopathologically. Armadillos with a lepromatous leprosy response were infected intravenously in the saphenous vein with 1x109 viable *M. leprae* derived from athymic nude mice. The NHDP armadillo colony is generally composed of ~60% *M. leprae*-infected armadillos and ~40% naïve armadillos at any given time. Experimental inoculations are done every 4 months; therefore, there are armadillos progressing at different leprosy stages throughout the colony.

Tissues were collected from the *M. leprae*-infected armadillos and used for the production of leprosy research reagents (*Larsen et al., 2020*). Four hours prior to sacrifice, the animal was given

Gentamicin and Penicillin. The armadillo was anesthetized using a combination of Ketamine (0.6 ml) and Dexdomitor (0.4 ml) given IM, and the skin of the abdomen was shaved and cleaned with iodine, 70% ethanol, and sterile water. The animal was euthanized by exsanguination and placed on its back in a BSC where the tissues were aseptically removed. The tissues were placed in sterile jars and, after checking for contaminants, stored at 70°C.

## Cancer data collection from previous studies

Published cancer prevalence data for from giant anteater (*M. tridactyla*), Southern tamandua (*T. tetradactyla*), silky anteater (*Cyclopes* sp.), maned three-toed sloths (*Bradypus torquatus*), brown-throated sloths (*B. variegatus*), pale-throated sloth (*Bradypus tridactylus*), Linné's two-toed sloth (*C. didactylus*), three-toed sloths (*Bradypus sp.*), and two-toed slots (*Choloepus sp.*) from four published surveys of morbidity and mortality in these species (*Arenales et al., 2020a*; *Arenales et al., 2020b*; *Diniz and Oliveira, 1999*; *Diniz et al., 1995*). Neoplasia prevalence in these species of *Xenarthra* were compared to other *Therian* mammals using data from two published studies that included pathology reports from 37 (*Boddy et al., 2020*) and 191 (*Vincze et al., 2022*) species of *Therian* mammals, as well as 3 species of cetaceans (*Albuquerque et al., 2018*). The total data set includes cancer prevalence data from 221 species. Neoplasia reports from *D. novemcinctus* from *Boddy et al., 2020* were combined with our new cancer prevalence data from the NHDP colony and *C. didactylus* data from *Vincze et al., 2022* were combined with *C. didactylus* data from *Arenales et al., 2020b*. CIs (95%) on lifetime neoplasia prevalence were estimated in PropCI package (https://github.com/shearer/PropCIs, *Vazquez, 2018*) in R using the Clopper-Pearson exact CI function: exactci(x, n, conf.level=0.95), where x is the number of successes (necropsies with neoplasia), n is the total sample size (number of necropsies), and conf.level=0.95 is the 95% lower and upper CIs.

## Acknowledgements

The authors would like to thank R Stevenson for collecting cancer incidence data from NHDP necropsy records and the San Diego Frozen Zoo for providing *Xenarthran* and elephant cell lines. The armadillo colony is supported by the NIH National Institute of Allergy and Infectious Diseases through an inter-agency agreement (No. AAI15006) with HRSA/HSB/NHDP, this study was supported by a National Institutes of Health (NIH) grant to VJL (R56AG071860) and a National Science Foundation (NSF) IOS joint collaborative award to VJL (2028459).

## Additional information

### Funding

| Funder | Grant reference number | Author |
|---|---|---|
| Division of Intramural Research, National Institute of Allergy and Infectious Diseases | AAI15006 | Maria T Pena Linda B Adams |
| National Institutes of Health | R56AG071860 | Vincent J Lynch |
| National Science Foundation | 2028459 | Vincent J Lynch |

The funders had no role in study design, data collection and interpretation, or the decision to submit the work for publication.

### Author contributions

Juan Manuel Vazquez, Conceptualization, Data curation, Formal analysis, Validation, Investigation, Visualization, Methodology, Writing - original draft, Writing - review and editing; Maria T Pena, Baaqeyah Muhammad, Morgan Kraft, Data curation; Linda B Adams, Vincent J Lynch, Conceptualization, Data curation, Formal analysis, Supervision, Funding acquisition, Validation, Investigation, Visualization, Methodology, Writing - original draft, Project administration, Writing - review and editing

## Author ORCIDs

Juan Manuel Vazquez (iD) http://orcid.org/0000-0001-8341-2390
Vincent J Lynch (iD) http://orcid.org/0000-0001-5311-3824

## Decision letter and Author response

Decision letter https://doi.org/10.7554/eLife.82558.sa1
Author response https://doi.org/10.7554/eLife.82558.sa2

---

## Additional files

### Supplementary files

• MDAR checklist

### Data availability

All data generated or analysed during this study are included in the manuscript, supporting files, and data.

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
