## [Editor Report]

This study is a useful extension of previous work on the relationship between body size and cancer risk and the mechanisms by which large-bodied mammals reduce their cancer risk. Through solid analyses of the genomes and several aspects of the cell biology of sloths, armadillos and their relatives, the study explores whether the evolution of large body size in their relatives (some extinct) was correlated with genomic changes such as the duplication of tumor suppressor genes, experimentally demonstrating that cells of Xenarthrans (sloths, armadillos, anteaters) are exceptionally sensitive to DNA damage. The study concerns a topic of great interest and contributes to our understanding of how cancer risk has evolved in mammals.

---

## [Decision Letter]

**Decision letter after peer review:**

Thank you for submitting your article "Parallel evolution of reduced cancer risk and tumor suppressor duplications in Xenarthra" for consideration by *eLife*. Your article has been reviewed by 2 peer reviewers, and the evaluation has been overseen by a Reviewing Editor and Patricia Wittkopp as the Senior Editor. The following individual involved in review of your submission has agreed to reveal their identity: Stephen C Stearns (Reviewer #1).

Essential revisions:

1) Please remove the section on cell doubling times or add data from more than one sloth.

2) Both reviewers have raised numerous points associated with statements made in the manuscript. It will be essential that the authors carefully address these and revise their text accordingly.

3) Both reviewers noted the typographical errors and the need for additional, careful editing. Please ensure your manuscript is carefully edited prior to resubmission.

*Reviewer #1 (Recommendations for the authors):*

The best way to strengthen this paper is to cut out the section on cell doubling times and wait to publish that until you can get data on more than one sloth.

I have three general questions that you might want to address in the introduction or discussion:

Is there any evidence from other lineages with reduced cancer risk that the gene duplications that reduce cancer risk first evolved in small animals, thus enabling the evolution of large body size, or do we not yet know the timing of those two events to say with some certainty that one preceded the other? I think you have something to say about that from your work on Proboscideans.

Does the fact that a gene is involved in cancer biology mean that it does not have other functions that might be the reason it was duplicated, which would make its association with cancer risk a by-product rather than a cause? Is there a way to evaluate this alternative using comparisons in genomic data? Unless you can answer this question with some logical rigor, you will give the impression that you are trying to confirm a favorite idea rather than test it objectively.

What are the costs of the duplicated genes in a small mammal that is at less cancer risk? Does the fact that they have not been lost not indicate that they may have important functions other than protecting against cancer, and, if that is the case, what does that do to your argument? Could it mean that the duplications were preserved because they had benefits unrelated to cancer?

And one specific question whose answer would strengthen your conclusions:

Are Xenarthrans enriched in genes and gene duplications that function specifically in sensitivity to DNA damage? That info is hard to pull out of the word cloud in Figure 3B.

On Page 5, you state that the duplications in the cingulate lineage were only enriched in a pathway unrelated to cancer biology, and then in the next sentence you state that armadillos have extra copies of tumor suppressor genes. Those two statements appear to be contradictory. Clarify.

On Page 34, let me suggest that a fourth panel in Figure 4 in which you plot reconstructed body size against reconstructed cancer risk to see whether Peto's paradox extends to extinct species would be interesting. It would of course potentially be subject to the criticism of circular reasoning given methods of ancestral state reconstruction, so you should address that issue.

*Reviewer #2 (Recommendations for the authors):*

This is a paper on a topic of great interest. However, I almost feel it was submitted before the last round of editing, because there seem to be a fair number of textual problems and some confusion in explanations.

First, I am not sure I understand the statement on page 5: "…notable decreases in intrinsic cancer risk in the Xenarthran stem-lineage, giant armadillo,…" The authors' intrinsic risk statistic depends only on body mass and life span. So if they reconstruct a reduced cancer risk, that must be due to reduced body size or lifespan inferred on that branch. Hence, I do not understand how they could infer a reduced risk for giant ground sloth along a branch with a massive increase in body size. Is the "K" statistic taken on the maximum of body size and lifespan over all of the species or computed for each species individually? My expectation is that the large, long lived species would have a larger intrinsic cancer risk given their body size/lifespan that is compensated for by the gene duplicates and so forth the authors compute. Referring to the Methods here does not clarify very much, since there are several subscripted K values used in ways that are not clearly defined.

Ideally, I would like to see an analysis of a control lineage's pattern of tumor suppressor duplications-I am not sure if perhaps the authors did this in their previous paper and could refer to that.

I also think the authors use of the phrase "sensitive to DNA damage" is a bit poorly chosen. To me, it implies that DNA damage is unusually likely to induce cancer or suchlike. Perhaps "intolerant to DNA damage" or something like this would be better?

---

## [Author Response]

Essential revisions:1) Please remove the section on cell doubling times or add data from more than one sloth.

We note that we have doubling time data for two species of sloth: Linnaeus's two-toed sloth (*Choloepus didactylus*) and Hoffmann's two-toed sloth (*Choloepus hoffmanni*). We would like to have additional species, but we are only aware of cell lines from these species. Therefore, we have not removed this part of the manuscript because we have more than one species.

2) Both reviewers have raised numerous points associated with statements made in the manuscript. It will be essential that the authors carefully address these and revise their text accordingly.

We have addressed these concerns, which we elaborate on below.

3) Both reviewers noted the typographical errors and the need for additional, careful editing. Please ensure your manuscript is carefully edited prior to resubmission.

We have carefully edited the manuscript and corrected typographical errors.

Reviewer #1 (Recommendations for the authors):The best way to strengthen this paper is to cut out the section on cell doubling times and wait to publish that until you can get data on more than one sloth.

We note that we have doubling time data for two species of sloth: Linnaeus's two-toed sloth (*Choloepus didactylus*) and Hoffmann's two-toed sloth (*Choloepus hoffmanni*). We would like to have additional species, but we are only aware of cell lines from these species. Therefore, we have not removed this part of the manuscript because we have more than one species.We have clarified this fact in the text. Furthermore, we highlight instances where species which we tested have published population doubling times, and we note that our population doubling estimates are in line with these estimates.

I have three general questions that you might want to address in the introduction or discussion:Is there any evidence from other lineages with reduced cancer risk that the gene duplications that reduce cancer risk first evolved in small animals, thus enabling the evolution of large body size, or do we not yet know the timing of those two events to say with some certainty that one preceded the other? I think you have something to say about that from your work on Proboscideans.

Our studies in Proboscideans suggest that gene duplications that may contribute to the evolution of reduced cancer risk occurred before the evolution of large bodies (Vazquez 2021,Vazquez 2018,Sulak 2016). These data suggest that reduced cancer risk, or at least the gene duplications that allow for reduced cancer risk, first evolved in small Afrotherians, including small-bodied Proboscideans. Thus these duplications may have enabled the evolution of large body sizes rather than causation in the other direction, i.e., selection for large bodies followed by the evolution of an increased number of tumor suppressors. We have added this point to the Discussion section.

Does the fact that a gene is involved in cancer biology mean that it does not have other functions that might be the reason it was duplicated, which would make its association with cancer risk a by-product rather than a cause? Is there a way to evaluate this alternative using comparisons in genomic data? Unless you can answer this question with some logical rigor, you will give the impression that you are trying to confirm a favorite idea rather than test it objectively.

This is an important consideration. We have approached this study with the a priori hypothesis that gene duplication may have played a role in the evolution of reduced cancer rates in Xenarthra; We note that this hypothesis is informed from our previous studies (Vazquez 2021,Vazquez 2018,Sulak 2016), including the paper from which this study extends (Vazquez 2021). We do not assume, however, that the primary selective advantage of fixing or maintaining a duplicate gene is an anticancer function or even that most gene duplications will be associated with anticancer or prolongevity functions; Indeed, many are likely associated with traits unrelated to cancer or lifespan biology. We also agree that the association between a gene (duplication) and cancer risk may be a by-product rather than a cause, i.e., the duplicate gene may have a secondary effect on reducing cancer that is not the selected effect that preserved the duplicate. In this sense such a duplication may be considered an exaptation for the evolution of large body sizes. We now note this in the section “Ideas and speculation”.

What are the costs of the duplicated genes in a small mammal that is at less cancer risk? Does the fact that they have not been lost not indicate that they may have important functions other than protecting against cancer, and, if that is the case, what does that do to your argument? Could it mean that the duplications were preserved because they had benefits unrelated to cancer?

Most genes with tumor suppressor functions have roles in multiple processes outside of their anticancer functions such as promoting cell proliferation during development; For example, *TP53* is essential for female reproduction and plays a role in orchestrating uterine progesterone response and promoting implantation (Hu 2007). The costs of losing these genes would likely be high, including infertility in p53 knockout mice. Therefore, we expect that single copy versions of the duplicate genes we identified are maintained as single copies in most small-bodied animals. However, whether there is a cost to duplicating tumor suppressors, in both small- and large-bodied animals is an open question; We have suggested that there is likely a cost to having increased tumor suppressor copy number, in particular because most of these genes are maintained as a single copy in most species (Sulak 2016,Vazquez 2021). But the costs of these duplicated tumor suppressor genes is also difficult to infer from functional genomics data.

We do not believe that these observations invalidate our hypothesis that duplication of genes with tumor suppressor functions may have contributed to the evolution of reduce cancer rates in Xenarthra for at least three reasons: (1) We are careful to not claim that the only consequences of these duplications is on anticancer or proaging phenotypes; (2) We are careful to not make statements about the percent of duplications that are associated with anticancer or proaging phenotypes; and (3) Our hypothesis is that duplication of tumor suppressor genes occurred coincident with anticancer and prolongevity cellular phenotypes and reduced cancer rates in Xenarthra, which we observe (we acknowledge this is not a proper falsifiable hypothesis *sensu* Popper).

Regarding the loss of gene duplications, it is difficult to ascertain both the recency of the “need” for these additional copies of tumor suppressors during the evolution of large bodies, as well as rate of loss of these duplicates. The fragmentary nature of the fossil record hides the number of true gigantic ancestral species within each clade; thus, it is possible that the need to conserve these duplicates was more recent than we can show here. Similarly, the issues of genome quality and quantity for *Xenarthra* limit our ability to robustly quantify exact gene copy number, which limits in turn our ability to estimate the rate of gene duplication losses. A true estimation of the rate of loss of tumor suppressor genes, and thus of the possible costliness of these duplicates, is therefore outside of the scope of this paper and of the field until newer, chromosome-level genomes are released for these and additional species.

And one specific question whose answer would strengthen your conclusions:Are Xenarthrans enriched in genes and gene duplications that function specifically in sensitivity to DNA damage? That info is hard to pull out of the word cloud in Figure 3B.

This is a very good point that we should have addressed in more detail. While most genes duplicated in the Xenarthran stem-lineage are related to cell cycle regulation, there are several enriched terms related to DNA damage and apoptotic responses including such as “Intrinsic Pathway for Apoptosis” (enrichment ratio = 6.76, hypergeometric P=9.75x10^-3^, FDR q=0.20), “Inhibition of replication initiation of damaged DNA by RB1/E2F1” (enrichment ratio = 15.26, hypergeometric P=7.30x10^-3^, FDR q=0.17), “Transcriptional Regulation by TP53” (enrichment ratio = 2.76, hypergeometric P=3.22x10^-3^, FDR q=0.11), and “Regulation of TP53 Degradation” (enrichment ratio = 8.50, hypergeometric P=5.14x10^-3^, FDR q=0.14). We have added this to the “Xenarthran cells are very sensitive to DNA damage” section of the manuscript. Several of these terms are also shown in the word cloud in Figure 3B.

On Page 5, you state that the duplications in the cingulate lineage were only enriched in a pathway unrelated to cancer biology, and then in the next sentence you state that armadillos have extra copies of tumor suppressor genes. Those two statements appear to be contradictory. Clarify.

We apologize for the confusion. We meant that armadillos have extra tumor suppressor genes because these genes duplication in the Xenarthran stem-lineage, rather than *additional* gene duplications in Cingulates. We have rephrased this sentence to read:

“These observations are consistent with earlier studies suggesting that Xenarthrans have extra copies of tumor suppressor genes (Tollis et al., 2020; Vazquez and Lynch, 2021)”

On Page 34, let me suggest that a fourth panel in Figure 4 in which you plot reconstructed body size against reconstructed cancer risk to see whether Peto's paradox extends to extinct species would be interesting. It would of course potentially be subject to the criticism of circular reasoning given methods of ancestral state reconstruction, so you should address that issue.

We have carefully considered this suggestion, but have decided to not include an additional panel in figure 4 with these data because they are already shown in Figure 2A. However, we did not previously indicate which lineages shown in Figure 2A were extinct. We have updated the figure to show which species are extinct with skull-and-crossbone icons as in Figure 1.

Reviewer #2 (Recommendations for the authors):This is a paper on a topic of great interest. However, I almost feel it was submitted before the last round of editing, because there seem to be a fair number of textual problems and some confusion in explanations.First, I am not sure I understand the statement on page 5: "…notable decreases in intrinsic cancer risk in the Xenarthran stem-lineage, giant armadillo,…" The authors' intrinsic risk statistic depends only on body mass and life span. So if they reconstruct a reduced cancer risk, that must be due to reduced body size or lifespan inferred on that branch. Hence, I do not understand how they could infer a reduced risk for giant ground sloth along a branch with a massive increase in body size. Is the "K" statistic taken on the maximum of body size and lifespan over all of the species or computed for each species individually? My expectation is that the large, long lived species would have a larger intrinsic cancer risk given their body size/lifespan that is compensated for by the gene duplicates and so forth the authors compute. Referring to the Methods here does not clarify very much, since there are several subscripted K values used in ways that are not clearly defined.

I (VJL) think this confusion results from a matter of perspective. It is the case that large, long lived species should have a larger intrinsic cancer risk given their body size/lifespan. However, the empirical observation is that cancer prevalence across broad taxonomic groups is relatively the same. Thus, we express cancer risk as the reduction in intrinsic risk that must have occurred for empirical cancer prevalence to be more or similar across species. We note that because the change in intrinsic risk data is expressed as log(δ K), one can easily reverse the sign to get the magnitude of intrinsic risk increase associated with body size and lifespan rather than the decrease necessary to keep observed cancer prevalence across species more or less similar.

Ideally, I would like to see an analysis of a control lineage's pattern of tumor suppressor duplications-I am not sure if perhaps the authors did this in their previous paper and could refer to that.

Indeed, this analysis is part of the paper on which the current study extends (Vazquez 2021). Briefly, while many lineages have duplication of tumor suppressor genes, the great over-representation of tumor suppressor duplications is only observed in lineages that evolved large bodies and long lifespans (Vazquez 2021).

I also think the authors use of the phrase "sensitive to DNA damage" is a bit poorly chosen. To me, it implies that DNA damage is unusually likely to induce cancer or suchlike. Perhaps "intolerant to DNA damage" or something like this would be better?

We agree that there is some ambiguity around the meaning of “sensitive”. However, this is intentional because we did not quantify DNA damage levels induced by MMC. Thus, our results can be explained by at least two, non-mutually exclusive mechanisms: (1) A derived DNA damage response is unusually likely to induce apoptosis in response to similar levels of DNA damage across species; or (2) Xenarthran cells accumulating more DNA damage than other species at equivalent doses of MMC. One of these could be considered to describe sensitivity while the other tolerance. Because we have not determined which mechanism is responsible for the MMC response difference across species we have opted to keep the term sensitivity (which is also consistent with our prior publications on this topic).